# TDSD: Text-Driven Scene-Decoupled Weakly Supervised Video Anomaly Detection

Shengyang Sun[*]
Zhejiang University
Hangzhou, Zhejiang, China
sunshy@zju.edu.cn

Jiashen Hua
Alibaba Cloud
Hangzhou, Zhejiang, China
jiashen.hjs@alibaba-inc.com

Junyi Feng
Alibaba Cloud
Hangzhou, Zhejiang, China
felix.fjy@alibaba-inc.com

Dongxu Wei
Alibaba Cloud
Hangzhou, Zhejiang, China
weidongxu.wdx@alibaba-inc.com

Baisheng Lai
Alibaba Cloud
Hangzhou, Zhejiang, China
baisheng.lbs@alibaba-inc.com

Xiaojin Gong[†]
Zhejiang University
Hangzhou, Zhejiang, China
gongxj@zju.edu.cn

## ABSTRACT

Video anomaly detection has garnered widespread attention in industry and academia in recent years due to its significant role in public security. However, many existing methods overlook the influence of scenes on anomaly detection. These methods simply label the occurrence of certain actions or objects as anomalous. In reality, scene context plays a crucial role in determining anomalies. For example, running on a highway is anomalous, while running on a playground is normal. Therefore, understanding the scene is essential for effective anomaly detection. In this work, we aim to address the challenge of scene-dependent weakly supervised video anomaly detection by decoupling scenes. Specifically, we propose a novel text-driven scene-decoupled (TDSD) framework, consisting of a TDSD module (TDSDM) and fine-grained visual augmentation (FVA) modules. The scene-decoupled module extracts semantic information from scenes, while the FVA module assists in fine-grained visual enhancement. We validate the effectiveness of our approach by constructing two scene-dependent datasets and achieve state-of-the-art results on scene-agnostic datasets as well. Code is available at https://github.com/shengyangsun/TDSD.

## CCS CONCEPTS

• **Computing methodologies → Scene anomaly detection**.

## KEYWORDS

Scene-Dependent Video Anomaly Detection, Weakly Supervised Learning, Text-Driven Scene-Decoupled.

[*]Work done during an internship at Alibaba Cloud.
[†]Corresponding Author.

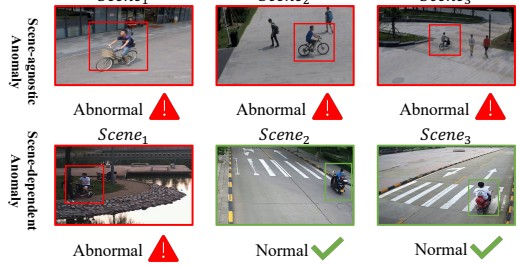

**(a) Scene-agnostic Anomaly vs. Scene-dependent Anomaly**

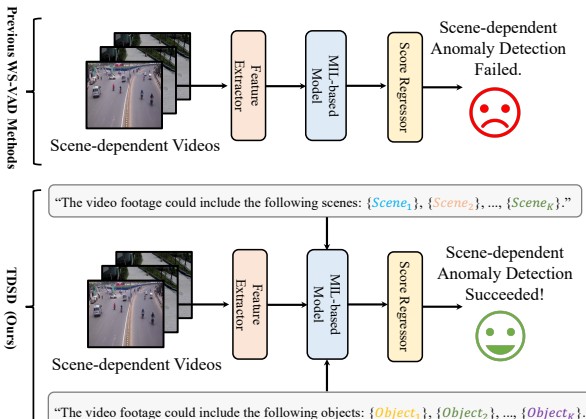

**(b) Previous WS-VAD methods vs. TDSD (Ours)**

**Figure 1: (a) Scene-agnostic anomalies refer to events that are independent of the scene in which they occur. However, scene-dependent anomalies require consideration of the scene in which the event occurs. (b) Previous weakly supervised methods were unable to detect scene-dependent anomalies, but by injecting semantic information about scenes, we can achieve the detection of scene-dependent anomalies.**

**ACM Reference Format:**
Shengyang Sun, Jiashen Hua, Junyi Feng, Dongxu Wei, Baisheng Lai, and Xiaojin Gong. 2024. TDSD: Text-Driven Scene-Decoupled Weakly Supervised Video Anomaly Detection. In *Proceedings of the 32nd ACM International Conference on Multimedia (MM '24), October 28-November 1, 2024, Melbourne, VIC, Australia.* ACM, New York, NY, USA, 10 pages. https://doi.org/10.1145/3664647.3680934

# 1 INTRODUCTION

Video anomaly detection (VAD) [1, 17, 23, 24, 31, 42, 56] aims to discriminate the abnormal frames from the given video, has received significant research interest due to its crucial role in miscellaneous domains such as public security and surveillance. Some previous methods [15, 18] explored training models in a fully-supervised setting, however, full supervision requires frame-level or snippet-level annotations for training, which entails a significant amount of manual labeling costs. Conversely, some researchers utilize unsupervised methods [8, 22, 24, 28] to address this task, only using normal videos to train the model and identifying outliers as anomalies during evaluation. Unfortunately, collecting all normal samples is impractical, and since no abnormal samples are involved in training, the model is prone to producing false alarms during inference. To reduce the cost of data labeling and obtain models with excellent performance, weakly supervised video anomaly detection (WS-VAD) [5, 7, 16, 29, 35, 42, 45, 54, 56] using only video-level annotations for training has recently received widespread attention.

In the task of anomaly detection, defining what constitutes an anomaly is crucial. Previous research has primarily focused on a single-scene scenarios [1, 17, 23, 31] or multi-scene scenarios [24, 42, 56] where anomalies exhibit consistent behavior across different scenes, known as scene-agnostic anomalies. However, in the real world, determining whether an event is anomalous often requires considering the scene context in which it occurs, *i.e.* scene-dependent anomalies, also known as scene-aware anomalies. As shown in Figure 1(a), it is abnormal for bicycles to travel on pedestrian pathways, whereas it is normal for them to travel on roads. By taking into account the scene context, anomaly detection systems can adapt to different environments, enhancing their scalability and applicability in real-world deployment. Therefore, in recent years, there have been several notable attempts at scene-dependent VAD [2, 3, 36, 38]. However, these studies have been conducted in unsupervised setting, which have limited performance. To enhance the practical application capabilities of VAD, in this work, we investigate scene-dependent anomaly detection under a weakly supervised setting.

In this work, our focus lies in the meticulous exploration of scene context within videos, enabling the model to possess the capability of detecting scene-dependent anomalies. To better describe the scene information in the video, we classify scene information into context descriptions and object descriptions. Specifically, the context descriptions encompass general scene summaries, such as *schoolhouse*, *palace*, and *store* for general descriptions, which is a macroscopic description of the overall scene. However, solely relying on macroscopic descriptions may make it difficult to distinguish between some similar scenes, such as park scenes and front yards of houses. Determining whether it is a park often requires considering the presence of *park benches* and *fountains*. Therefore, when exploring scene information, we simultaneously consider the objects of the scene, such as *bicycle*, *van*, and *fountain* for specific objects. Exploiting scene objects not only enhances the scene's discriminative ability but also incorporates potential abnormal targets, thereby increasing the anomaly detection capability of the model.

In real-world VAD tasks, scenes are often complex, and it's difficult to fully capture the semantic meaning of video segments

solely through visual features [4]. In recent years, the emergence of vision-language models, *e.g.* CLIP [30] and ALIGN [12], has made it possible to combine visual information from videos with natural language descriptions, obtaining richer information from two modalities. This fusion provides a more comprehensive understanding of scenes, thereby aiding in more accurate anomaly detection. Based on this, we propose a text-driven scene-decoupled (TDSD) framework for weakly supervised video anomaly detection, which exploits the context descriptions and object descriptions of scenes based on the pre-trained CLIP model, empowering the model to detect scene-dependent anomalies, as shown in Figure 1(b).

Our contributions are summarized as follows:

- We propose a novel text-driven scene-decoupled framework to address weakly supervised video anomaly detection, which exploits the context and objects semantic meanings of scenes in normal and abnormal videos, enabling the model to detect scene-dependent anomalies. To the best of our knowledge, this is the first work to address scene-dependent video anomaly detection under a weakly supervised setting.
- We designed the text-driven scene-decouple module (TDSDM) and the fine-grained visual augmentation (FVA) module for extracting semantic features of scenes and enhancing features at a fine granularity, respectively. The TDSDM consists of context semantic injection (CSI) and object semantic injection (OSI), which enable the model to learn the semantic features of scenes and objects within scenes, respectively.
- To validate the ability to detect scene-dependent anomalies of our method, we reorganize the scene-dependent anomaly dataset NWPU Campus [3], which is originally constructed for the one-class classification setting, to suit the weakly supervised setting. Besides, we merged the public scene-agnostic datasets UCF-Crime and ShanghaiTech into one scene-dependent dataset, further validating the effectiveness of the WS-VAD method. Additionally, we conduct experiments on two scene-agnostic datasets to validate the aspects of our designs that are independent of scene awareness.

## 2 RELATED WORKS

### 2.1 Video Anomaly Detection

Video anomaly detection (VAD) has gained significant research interest, which can be classified three main research lines, fully-supervised [15, 18], unsupervised [10, 19, 22, 24, 27, 28, 37, 44, 50, 53] and weakly-supervised VAD [5–7, 16, 29, 35, 39–42, 45, 54, 56].

The fully supervised VAD [15, 18] trains a model using frame-level annotations, including precise bounding boxes of anomalies. This necessitates a significant amount of labor to label the data. To alleviate the burden of annotating numerous samples, unsupervised methods focus solely on gathering normal videos for model training. During inference, they differentiate samples deviating from normality as anomalies. For example, reconstruction-based unsupervised techniques [8, 22, 24, 28] employ autoencoders to encode normal samples into latent spaces, identifying poorly reconstructed samples as anomalies. For instance, Yu *et al.* [50] propose a reconstruction-based method that uses an object detector to identify potential anomalies in videos and focuses on the

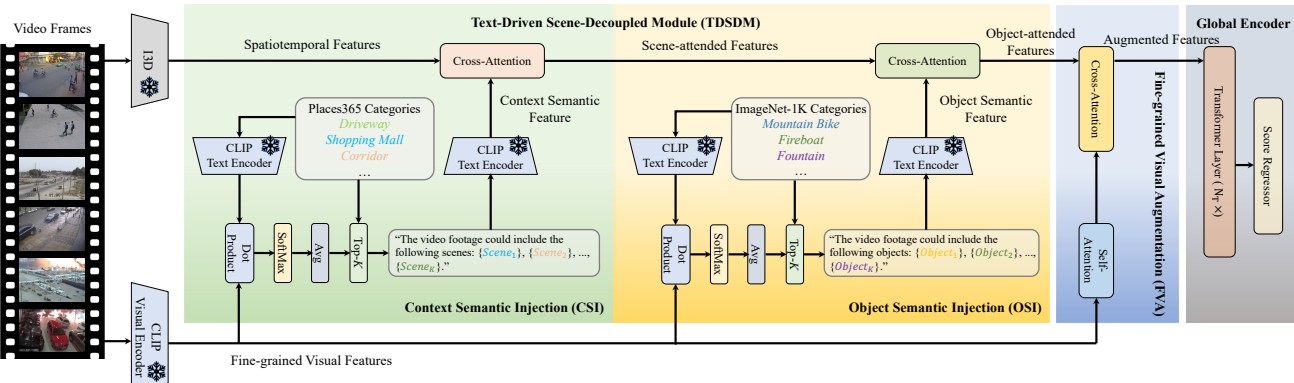

**Figure 2: An overview of the proposed framework. It includes the text-driven scene-decouple module (TDSDM), fine-grained visual augmentation (FVA), and the global encoder. The snowflake icon in the figure indicates that we have frozen this module in the training. Best viewed in color.**

temporal information of objects. Meanwhile, distance-based unsupervised methods [10, 11, 34, 37] establish decision boundaries using Gaussian mixture models or one-class SVMs, distinguishing deviated data as anomalous samples. However, since abnormal samples are absent from the training set, the model tends to generate false alarms when detecting ambiguous normality. To alleviate the burden of labor-intensive annotations while training models on abnormal data, weakly-supervised video anomaly detection (WS-VAD) [7, 29, 35, 42] tackles this challenge by employing multiple instance learning (MIL). This approach trains the model using video-level annotated data and achieves excellent performance. For example, Lv *et al.* [25] propose an unbiased multiple instance learning (UMIL) framework to tackle bias in MIL-based VAD, thereby decreasing false alarms and enhancing performance. Wu *et al.* [48] propose a framework that leverages the pre-trained vision-language model CLIP to address the WS-VAD task. All previous weakly supervised methods focus on scene-agnostic anomaly detection, leaving scene-dependent VAD unexplored. In contrast to them, our approach focuses on addressing both scene-agnostic anomalies and exploring solutions for scene-dependent anomalies.

## 2.2 Scene-dependent Video Anomaly Detection

In recent years, scene-dependent VAD has garnered increasing attention due to the growing complexity of surveillance scenarios. The incorporation of scene context into VAD involves several crucial components, such as extracting scene features, identifying scene types, and modeling the relationship between objects and scenes. In previous methods, scene features have been extracted by inputting the entire frame, with or without marked objects, into various encoders [2, 3, 36, 38]. Scene types have been identified primarily using unsupervised clustering algorithms [2, 36, 38]. Additionally, spatiotemporal context graph [36], hierarchical scene normality-binding model [2], and scene-conditioned variational auto-encoder [3] are constructed to model the relationship between objects and scenes. In addition, several studies have employed graphs to model scenes and objects. Liu *et al.* [20] employ three models to construct a factual causal graph for video action recognition. Han *et al.* [9] propose a method that improves the accuracy of scene graph generation by

decomposing the predicate prediction task into subtasks. In contrast, we leverage the pre-trained vision-language model to obtain the scene semantic features and objects semantic features, injecting the semantic features into the spatiotemporal features, being the scene-attended features. Besides, this is the first work to address the scene-dependent VAD under a weakly supervised setting.

## 2.3 Vision-language Models

Vision-language models have been widely applied to various vision tasks. For instance, Liu *et al.* [21] propose a multi-grained gradual inference model that focuses on objects and aligns textual and visual information through graph structures and its multi-grained gradual inference mechanism. Besides, pre-trained vision-language models (VLM) such as CLIP [30] and ALIGN [12], which are trained on large-scale datasets, have acquired a wide range of knowledge and demonstrated impressive generalization capabilities. Therefore, in recent years, these models have been extensively deployed to downstream vision tasks through adaptation techniques like fine-tuning and prompt learning [13, 32, 49, 57]. In the context of weakly supervised VAD, Joo *et al.* [14] leverage the CLIP vision encoder along with a proposed temporal self-attention module to enhance feature quality. Wu *et al.* [48] adapt CLIP by incorporating an adaptation layer into the CLIP vision encoder and employing a text prompt learning scheme for the CLIP text encoder. Zanella *et al.* [51] directly manipulate the latent CLIP feature space to identify the normal event subspace. In contrast, our approach utilizes CLIP's powerful zero-shot capability to obtain semantic information about scenes. By employing CLIP's text encoder, we extract semantic features of scenes, which are then integrated with spatiotemporal features. This enables the model to detect scene-dependent anomalies.

## 3 METHODOLOGY

## 3.1 Overview of the Proposed Framework

We illustrate the proposed framework in Figure 2. The framework mainly includes three parts, the text-driven scene-decoupled module (TDSDM) including the context semantic injection (CSI) and the object semantic injection (OSI), the fine-grained vision augmentation (FVA), and the last part is the global encoder that consists of

a $N_T$-layer Transformer and a MLP-based regressor. The TDSDM is used to decouple scenes. The CSI module provides a general description of the scene and generates semantic features of the scene, while the OSI module provides descriptions of objects within the scene and generates semantic features of the objects in the scene. The FVA module provides fine-grained visual augmentation, which involves enhancing a snippet-level spatiotemporal feature with $T$ frame-level CLIP features. The augmented features finally pass through a global encoder to aggregate context overall snippets in a video and then predict anomaly scores.

## 3.2 Spatiotemporal Features Extraction

The spatiotemporal features extraction aims to extract the one spatiotemporal feature for each video snippet that contains $T$ consecutive frames (here $T = 16$ following [5, 7, 35, 42, 56]). Specifically, given the $i$-th video snippet of dimension $T \times H \times W \times C$ is fed into the pre-trained I3D feature extractor that fine-tuned on Kinetics-400 dataset, extracting the spatiotemporal features $\dot{\mathbf{F}}_i^{I3D} \in \mathbb{R}^{D_I}$, where the $H$, $W$ and $C$ are the height, width and channel of each frame, respectively, and $D_I$ is the dimension of extracted features. Then the extracted features pass through a linear project layer $f_{I3D}(\cdot) : \mathbb{R}^{D_I} \rightarrow \mathbb{R}^D$ to produce the $\mathbf{F}_i^{I3D} \in \mathbb{R}^D$ for fitting the subsequent modules. Following [5, 7, 35, 42, 56], we divide each video into 32 snippets by conducting the average temporal pooling in the training stage.

## 3.3 Text-Driven Scene-Decoupled Module

### 3.3.1 *Context Semantic Injection (CSI)*. The CSI module is used to obtain a general description of the scene and to fuse the semantic features of the context with the spatiotemporal features. With the help of the powerful detection ability of the CLIP, we use this VLM model to detect the different scenarios in zero-shot for each video snippet. Firstly, the 365 scene categories borrowed from the Places365 dataset [55] are fed into the CLIP's text encoder, achieving the scene weight matrice:

$$\mathcal{W}^S = \mathcal{E}_T([scene_1, scene_2, ..., scene_{365}]), \quad (1)$$

where the $\mathcal{W}^S \in \mathbb{R}^{365 \times D_C}$ indicates the weight matrices of scenes, the $\mathcal{E}_T(\cdot)$ denotes the text encoder of CLIP, and the $scene_i$ denotes the $i$-th category of Places365, *e.g.* schoolhouse, shopping mall, parking garage, *etc.* Then we feed the $T$ frames into the CLIP visual encoder for each video snippet to obtain the visual representations:

$$\mathbf{F}_i^{CLIP} = \mathcal{E}_V([\mathbf{I}_{i,1}, \mathbf{I}_{i,2}, ..., \mathbf{I}_{i,T}]), \quad (2)$$

where the $\mathbf{F}_i^{CLIP} \in \mathbb{R}^{T \times D_C}$ are the visual features of $i$-th snippet produced by the CLIP, the $\mathcal{E}_T(\cdot)$ denotes the visual encoder of CLIP, $\mathbf{I}_{i,j} \in \mathbb{R}^{C \times H \times W}$ denotes the $j$-th frame of the $i$-th snippet. Thus, the matching probabilities between each frame and the scene categories can be obtained by

$$\mathbf{P}_i^{Scene} = \text{SoftMax}(\|\mathbf{F}_i^{CLIP}\|_2 \otimes \|\mathcal{W}^S\|_2^T), \quad (3)$$

where the $\mathbf{P}_i^{Scene} \in \mathbb{R}^{T \times 365}$ denotes the matching probabilities of the $T$ frames of the $i$-th snippet, $\| \cdot \|_2$ is the $l_2$ norm, and the $\otimes$ represents the matrix multiplication operation. We average the probabilities across the $T$ frames and take the $K$ scene categories

which have the top-$K$ probabilities as the label scenarios:

$$[scene_1, scene_2, ..., scene_K] = \text{topK}(\text{Avg}(\mathbf{P}_i^{Scene})), \quad (4)$$

where the $\text{Avg}(\cdot) : \mathbb{R}^{T \times 365} \rightarrow \mathbb{R}^{365}$ denotes the average operation, and the $\text{topK}(\cdot)$ indicates that take the $K$ scene categories from the Places365 according to the top-$K$ probabilities. To obtain the semantic scene feature of $i$-th snippet, we put the $K$ scene categories into a guided language text $\mathcal{T}^S =$ *"The video footage could include the following scenes: {scene$_1$}, {scene$_2$}, ..., {scene$_K$}."*, then feeding the above text into the CLIP text encoder $\mathcal{E}_T(\cdot)$ to extract the context semantic feature $\mathbf{F}_i^{Scene} \in \mathbb{R}^D$:

$$\mathbf{F}_i^{Scene} = f_{scene}(\mathcal{E}_T(\mathcal{T}^S)), \quad (5)$$

where $f_{scene}(\cdot) : \mathbb{R}^{D_C} \rightarrow \mathbb{R}^D$ is the linear projection layer. Finally, both the spatiotemporal feature $\mathbf{F}_i^{I3D} \in \mathbb{R}^D$ and the scene clue feature $\mathbf{F}_i^{Scene}$ pass through the fusion layer to produce the scene-attended feature $\hat{\mathbf{F}}_i^{I3D} \in \mathbb{R}^D$, which is computed by the multi-head cross-attention followed by a feedforward (FFN) layer [43]:

$$\hat{\mathbf{F}}_i^{I3D} = \text{Cross-Attention}(\mathbf{F}_i^{I3D}, \mathbf{F}_i^{Scene}), \quad (6)$$

where the Cross-Attention$(\mathbf{x}, \mathbf{y})$ for each head computes dot-product attention as follows:

$$\begin{aligned} \text{Cross-Attention}(\mathbf{x}, \mathbf{y}) &= \text{Attention}(\mathbf{x}\mathbf{W}^Q, \mathbf{y}\mathbf{W}^K, \mathbf{y}\mathbf{W}^V) \\ &= \text{SoftMax}\left(\frac{\mathbf{x}\mathbf{W}^Q(\mathbf{y}\mathbf{W}^K)^T}{\sqrt{D}}\right)\mathbf{y}\mathbf{W}^V, \end{aligned} \quad (7)$$

where $\mathbf{W}^Q, \mathbf{W}^K, \mathbf{W}^Q \in \mathbb{R}^{D \times D_H}$ are learnable matrices with $D_H = D/N_H$, where $N_H$ is the number of attention heads.

### 3.3.2 *Object Semantic Injection (OSI)*. The OSI module is used to provide descriptions of objects within the scene and generate semantic features of the objects in the scene. Similar to the design of the CSI module, the OSI module powered by the CLIP detects objects in zero-shot mode. In detail, we feed 1,000 object categories borrowed from the ImageNet-1K [33] into the CLIP's text encoder to obtain the object weight matrix:

$$\mathcal{W}^O = \mathcal{E}_T([object_1, object_2, ..., object_{1000}]), \quad (8)$$

where the $\mathcal{W}^O \in \mathbb{R}^{1000 \times D_C}$ indicates the weight matrices of objects. With the extracted fine-grained visual feature according to Equation (2), the match probabilities between each frame and the object classes are computed by

$$\mathbf{P}_i^{Object} = \text{SoftMax}(\|\mathbf{F}_i^{CLIP}\|_2 \otimes \|\mathcal{W}^O\|_2^T). \quad (9)$$

Similar to the Equation (4), the $K$ objects classes are obainted as follows:

$$[object_1, object_2, ..., object_K] = \text{topK}(\text{Avg}(\mathbf{P}_i^{Object})), \quad (10)$$

then these object categories are put into the text $\mathcal{T}^O =$ *"The video footage could include the following objects: {object$_1$}, {object$_2$}, ..., {object$_K$}."*, thus the object semantic feature $\mathbf{F}_i^{Object} \in \mathbb{R}^D$ for $i$-th snippet is extracted as follows,

$$\mathbf{F}_i^{Object} = f_{object}(\mathcal{E}_T(\mathcal{T}^O)), \quad (11)$$

where $f_{object}(\cdot) : \mathbb{R}^{D_C} \to \mathbb{R}^D$ is the linear projection layer. The object feature $\mathbf{F}_i^{Object}$ and the scene-attended feature $\hat{\mathbf{F}}_i^{I3D}$ are fused with using the cross-attention, being the object-attended feature:

$$\tilde{\mathbf{F}}_i^{I3D} = \text{Cross-Attention}(\hat{\mathbf{F}}_i^{I3D}, \mathbf{F}_i^{Object}). \tag{12}$$

## 3.4 Fine-grained Visual Augmentation (FVA)

The purpose of the FVA is to enhance a snippet-level spatiotemporal feature with $T$ frame-level CLIP features, thus enhancing the visual expression capability of the features. To this end, we first obtain the fine-grained feature $\hat{\mathbf{F}}_i^{CLIP}$ computed by the $\mathbf{F}_i^{CLIP}$ with the Self-Attention($\mathbf{F}_i^{CLIP}$):

$$\text{Self-Attention}(\mathbf{x}) = \text{Attention}(\mathbf{x}\mathbf{W}^Q, \mathbf{x}\mathbf{W}^K, \mathbf{x}\mathbf{W}^V), \tag{13}$$

where $\mathbf{W}^Q, \mathbf{W}^K, \mathbf{W}^Q$ are learnable matrices as the same to Equation (7).

Finally, we can achieve the augmented spatiotemporal feature $\bar{\mathbf{F}}_i^{I3D} \in \mathbb{R}^D$ that augmented by the frame-level fine-grained features as follows:

$$\bar{\mathbf{F}}_i^{I3D} = \text{Cross-Attention}(\tilde{\mathbf{F}}_i^{I3D}, \hat{\mathbf{F}}_i^{CLIP}). \tag{14}$$

## 3.5 Network Training

In the weakly supervised VAD task, each training video is annotated with a binary label $y \in \{0, 1\}$ to denote whether this video is an abnormal video or not. When the enhanced features of all snippets within one video are produced, we employ a $N_T$-layer Transformer to model the global context of all snippets, and then we use a regressor to predict the anomaly scores:

$$\mathbf{s} = \Theta(\text{Transformer}_{(N_T \times)}(\bar{\mathbf{F}}^{I3D})), \tag{15}$$

where $\mathbf{s} \in \mathbb{R}^{N_S}$ is the anomaly scores of all snippets within one video and $\Theta(\cdot) : \mathbb{R}^D \to \mathbb{R}$ is the regressor that is implemented by a three-layer multi-layer perceptron (MLP).

Following the previous works, we adopt the binary cross-entropy loss to train the model, which classifies a video into abnormal or normal classes. Specifically, we average the top-$N_S$ anomaly scores as follows:

$$\bar{s} = \frac{1}{N_S} \sum_{s_i \in \mathcal{T}_{N_S}(\mathbf{s})} s_i, \tag{16}$$

where $\mathcal{T}_{N_S}(\mathbf{s})$ indicates the set of top-$N_S$ scores in $\mathbf{s}$. Then the MIL loss is defined by

$$\mathcal{L}_{MIL} = -y\log(\bar{s}) - (1 - y)\log(1 - \bar{s}). \tag{17}$$

## 4 EXPERIMENTS

## 4.1 Datasets and Evaluation Metric

*4.1.1 Datasets.* We conduct experiments to fully evaluate our proposed framework based on public datasets for VAD task, ShanghaiTech [19], UCF-Crime [35] , TAD [26], XD-Violence(XD) [47] and NWPU campus [3]. The large-scale dataset NWPU is the only one that considers scene-dependent anomalies. It is originally designed for the unsupervised setting, including only normal videos for training. To adapt it to the weakly supervised setting, we reorganize the dataset by selecting both normal and abnormal videos for the training set. Additionally, we ensure that the anomalies present

**Table 1: The statistics of five datasets under the weakly-supervised setting, *i.e.* TAD, XD-Violence (XD), re-organized NWPU Campus (NWPU), and the UCF_SHT which merged the UCF-Crime and ShanghaiTech datasets. Whether a dataset is scene-dependent (SD) or not is also indicated.**

| Dataset | Training set | | Test set | | #Anomaly types | SD |
|---------|---------|---------|---------|---------|---------|-----|
| | #Frames | #Videos | #Frames | #Videos | | |
| TAD [26] | 449,292 | 400 | 88,052 | 100 | 7 | ✗ |
| XD [47] | 16,378,527 | 3,954 | 2,335,801 | 800 | 6 | ✗ |
| NWPU [3] | 905,532 | 316 | 560,541 | 231 | 28 | ✓ |
| UCF_SHT | 12,804,648 | 1,848 | 12,520,52 | 489 | 24 | ✓ |

in the test set are also included in the training set, which is a fundamental assumption in scene-dependent VAD. The reorganized split consists of 316 training videos and 231 test videos. More details are provided in the supplementary material. To further valid the performance of detecting the scene-dependent anomalies, we merge the two scene-agnostic datasets UCF-Crime and ShanghaiTech into one dataset (*i.e.* merging the two training/test sets into one training/test set), named UCF_SHT. Due to the anomaly type of the two datasets being different, *e.g.* the appearance of a car is an abnormal event in ShanghaiTech while that event is normal in UCF-Crime, therefore the anomalies in UCF_SHT are scene-dependent. The statistics of all datasets are presented in Table 1.

*4.1.2 Evaluation Metric.* For the performance evaluation, following the common practice [3, 5, 7, 19, 35, 42, 48, 56], we adopt the frame-level area under the ROC curve (AUC) as the evaluation metric for TAD, NWPU campus, and UCF_SHT datasets, and use the average precision (AP) for the XD-Violence dataset. A higher AUC or AP denotes better performance.

## 4.2 Implementation Details

The proposed method is implemented in Pytorch. We employ the pre-trained CLIP (ViT-L/14) backbone to extract the features of $\mathbf{F}_i^{CLIP}$, $\mathbf{F}_i^{Scene}$ and $\mathbf{F}_i^{Object}$. The dimension of feature $D$ is set to 512. The top-$K$ in Equation (4) and Equation (10) is set to 5. The number of Transformer layer $N_T$ is set to 2 on UCF_SHT, TAD, and XD-Violence datasets, and set to 5 for the NWPU dataset. The $N_S$ in Equation (16) is set to 3. We train the proposed model on one NVIDIA Tesla V100 GPU in an end-to-end manner using the SGD optimizer for 50 epochs, the weight decay is set to 0.0005, the batch size is 32 and the learning rate is set to 0.0005 for TAD, UCF_SHT, and XD-Violence datasets and 0.005 for NWPU dataset.

## 4.3 Ablation Studies

To fully validate the effectiveness of the proposed method, we carefully examine the design of each module in our framework and investigate the performance of various model variants on two scene-dependent datasets *i.e.* UCF_SHT and NWPU, and two scene-agnostic datasets, *i.e.* XD-Violence and TAD.

*4.3.1 Effectiveness of the Proposed Modules.* The major contribution modules in the proposed framework are text-driven scene-decoupled module (TDSDM) and fine-grained visual augmentation (FVA). Therefore, we first conduct ablation studies to investigate the effectiveness of two modules by leaving out either one or both

**Table 2: Ablation studies of variant components of our proposed framework.**

| Index | TDSDM | FVA | UCF_SHT (%) | NWPU (%) | XD (%) | TAD (%) |
|---|---|---|---|---|---|---|
| 1 | ✗ | ✗ | 79.92 | 69.73 | 74.86 | 87.63 |
| 2 | ✓ | ✗ | 85.36 | 77.90 | 83.31 | 90.06 |
| 3 | ✗ | ✓ | 80.94 | 70.01 | 76.68 | 88.57 |
| 4 | ✓ | ✓ | **85.94** | **80.22** | **84.69** | **93.90** |

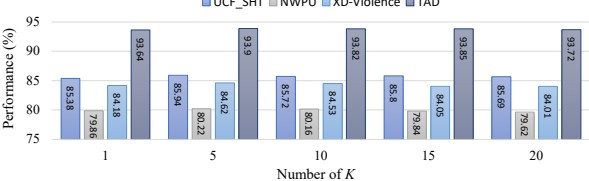

**Figure 3: The performance with a variant number of $K$ in the top-$K$ selection in CSI and OSI module. Best viewed in color.**

of them from our proposed framework. When the TDSDM and FVA both are removed, we only use the global encoder to produce the anomaly scores. Performances are reported in Table 2.

The results from the table show that solely employing the TDSDM or FVA both improves the performance. By adding to the TDSDM, we can observe that the improvements on scene-dependent datasets (UCF_SHT and NWPU) are more significant than the performances on the scene-agnostic datasets (XD-Violence and TAD), *e.g.* 69.73% *v.s.* 77.90% on NWPU compared with 87.63% *v.s.* 90.06% on TAD. For the FVA module, the results show that the performances have also been improved not only on the scene-dependent datasets but also on the scene-agnostic datasets. Besides, the synergy of TDSDM with FVA boosts performance significantly.

*4.3.2 **Effectiveness of the Designs in TDSDM**.* We then conduct experiments to study the effectiveness of the varied components in the core module of TDSDM. This module mainly has two components: 1) the context semantic injection (OSI) and 2) the object semantic injection (OSI). Each above component is validated in Table 3. The framework contains FVA and a global encoder when removing both two parts. The CSI is employed to obtain the general semantic meaning of the scene. From the results, we can observe that performance has significant improvements by considering the context semantic meaning, especially on two scene-dependent datasets, *i.e.* from the AUC of 80.94% to 85.61% on UCF_SHT and 70.01% to 78.05% on NWPU. The OSI module is mainly used to obtain the semantic meanings of the objects within the scene. With the help of the OSI (Index 3 in table), performances on all four datasets are improved, *e.g.* from 88.57% of AUC on TAD to 93.28%. In addition, the model achieves the best performance on all datasets by employing all components of TDSDM.

*4.3.3 **Impact of the hyperparameter top-$K$**.* To investigate the impact of the variant $K$ in Equation (4) and Equation (10) for selecting the top-$K$ scenes and top-$K$ objects, we conduct experiments that vary the $K$ from 1 to 20, reporting the results in Figure 3. Specifically, we choose the $K = 5$ because the model achieves the best performance. Additionally, we observe that the performance varies slights with different $K$, *e.g.* from 93.64% to 93.90% on TAD, which denotes the model is not sensitive to the $K$.

**Table 3: Ablation studies of variant components of the proposed TDSDM.**

| Index | CSI | OSI | UCF_SHT (%) | NWPU (%) | XD (%) | TAD (%) |
|---|---|---|---|---|---|---|
| 1 | ✗ | ✗ | 80.94 | 70.01 | 76.68 | 88.57 |
| 2 | ✓ | ✗ | 85.61 | 78.05 | 84.02 | 92.83 |
| 3 | ✗ | ✓ | 85.53 | 78.32 | 83.96 | 93.28 |
| 4 | ✓ | ✓ | **85.94** | **80.22** | **84.69** | **93.90** |

**Table 4: Ablation studies of variant CLIP backbones.**

| Index | Backbone | UCF_SHT (%) | NWPU (%) | XD (%) | TAD (%) |
|---|---|---|---|---|---|
| 1 | ViT-B/16 | 85.78 | 80.15 | 84.14 | 93.89 |
| 2 | ViT-B/32 | 85.43 | 80.01 | 84.27 | 94.49 |
| 3 | ViT-L/14 | **85.94** | **80.22** | **84.69** | 93.90 |

*4.3.4 **Impact of the Number of Layers in Global Encoder**.* We further conduct experiments to investigate the impact of the number of Transformer layers $N_T$ in the global encoder, which varies the $N_T$ from 1 to 6, and results are provided in Figure 5. From the figure, we observe that setting to 2 or 5 for the Transformer layers is sufficient, because the model achieves the best performance when the $N_T$ is set to 2 on the UCF_SHT, XD-Violence, and TAD, but the best setting is 5 for the NWPU dataset.

*4.3.5 **Impact of Variant CLIP backbones**.* The pre-trained CLIP backbone is employed in our framework to extract both fine-grained visual and text features. Thus we investigate the impact of the CLIP backbones, such as ViT-B/16, ViT-B/32, and ViT-L/14, results are reported in Table 4. From the results, we can observe that the model obtains the best performance when we select the ViT-L/14 as the backbone. Besides, the results show that the model can obtain the approximate performances by using the ViT-B/16 or ViT-B/32 compared with the ViT-L/14.

## 4.4 Qualitative Results

*4.4.1 **Predicted Anomaly Scores**.* We first visualize the anomaly scores predicted by the proposed framework on test sets of two scene-dependent datasets (UCF_SHT and NWPU) and two scene-agnostic datasets (XD-Violence and TAD), where a higher predicted score indicates a higher probability that the event is anomalous, results are shown in Figure 4.

For the evaluation of scene-dependent datasets, our proposed method effectively detects scene-dependent anomalous events. For example, in the scene-dependent dataset UCF_SHT, the appearance of a bicycle in the scene sh®®own in video *01_0139* is considered an anomalous event, while the appearance of a bicycle in the video *Normal_Videos_210* is considered as a normal event. Similarly, in the scene-dependent dataset NWPU, the appearance of a bicycle in the scene shown in video *D235_07* is considered an anomalous event, while the appearance of a bicycle in the scene shown in video *D031_09* is considered as a normal event. From the results in the figure, we can see that for the abnormal videos *01_0139* and *Normal_Videos_210*, our model produces high prediction scores for anomalous snippets, while producing low prediction scores for normal videos. In different scenes where the same event occurs, such as the appearance of a bicycle in the two scenes mentioned

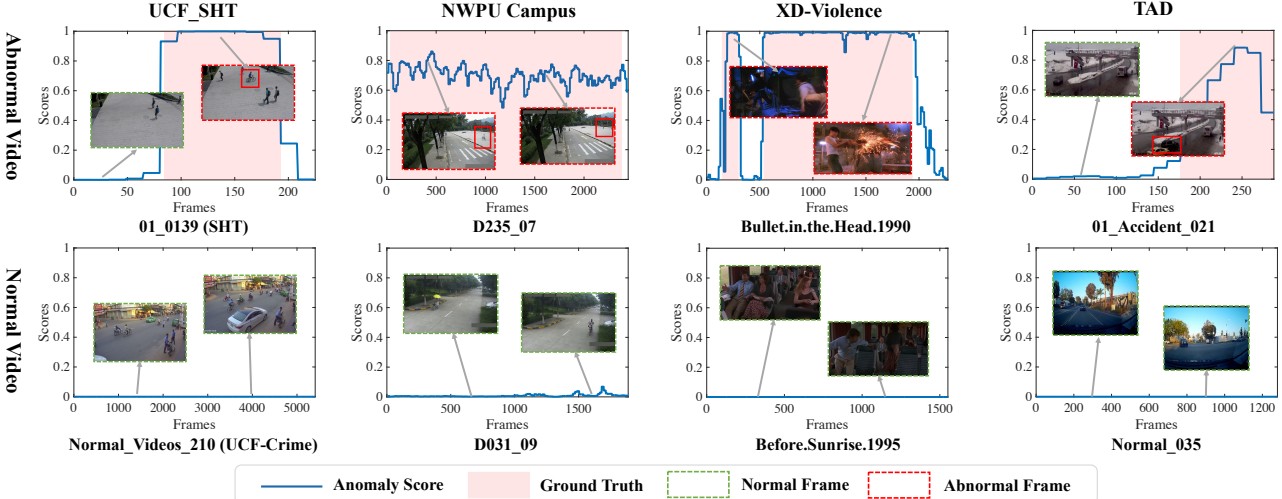

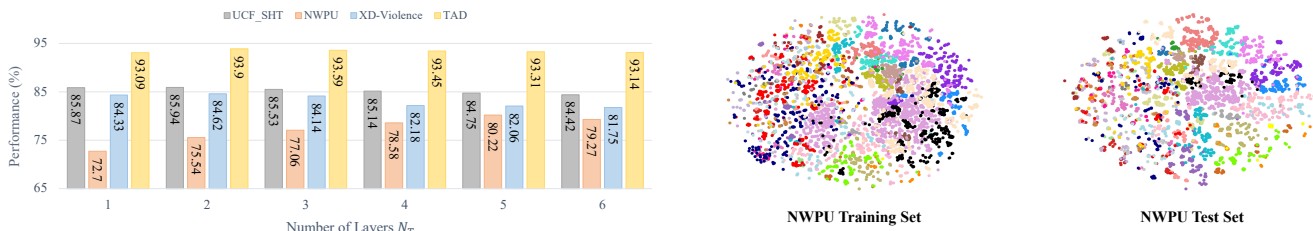

Figure 4: Visualization of anomaly scores predicted on the test sets of two scene-dependent datasets (UCF_SHT and NWPU) and two scene-agnostic datasets (XD-Violence and TAD). Best viewed in color.

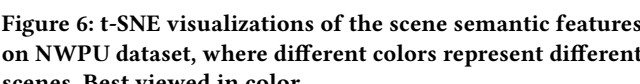

Figure 5: The performance with a different number of Transformer layers in the global encoder. Best viewed in color.

Figure 6: t-SNE visualizations of the scene semantic features on NWPU dataset, where different colors represent different scenes. Best viewed in color.

above, our model generates different anomaly scores. This indicates that our model can detect scene-dependent anomalous events.

For scene-agnostic datasets (XD-Violence and TAD), our model also accurately predicts anomalous events while generating low anomaly scores for normal events. Additionally, from the video *Bullet.in.the.Head.1990* in XD-Violence, it can be seen that our model can detect discontinuous anomalous events within a video.

*4.4.2 Generated Scene Text Results.* In the CSI module, we initially use CLIP for scene classification and then generate textual descriptions of scenes to obtain their semantic features. Consequently, we analyzed the generated scene text, *i.e.* the guided language text $\mathcal{T}^S$, as illustrated in Figure 7. We select scenes from four datasets: UCF_SHT, NWPU, XD-Violence, and TAD, and generate textual descriptions of these scenes. From the results of the generated text descriptions, it is evident that the descriptions of the scenes are fairly accurate. For instance, the first image depicts a scene of a store selected from UCF-Crime, and the third image depicts a scene of a campus selected from the NWPU dataset. Both the generated text accurately describes the scenes.

*4.4.3 Generated Object Text Results.* Similarly, in the OSI module, textual descriptions are primarily used to exploit objects within scenes, enhancing the representation capability of features. Therefore, we qualitatively analyze the precision of the generated text

for scene objects, as shown in Figure 8. The results from the figures show that the textual descriptions can capture the key objects within the scenes. For instance, the sliding door and the dog in the first scene, and various objects like motorcycles, bicycles, traffic lights, and aircraft in the other open scenes.

*4.4.4 Visualization of Scene Semantic Features.* After obtaining the generated textual descriptions of scenes $\mathcal{T}^S$, we used CLIP's text encoder to obtain semantic features of the scenes $\mathbf{F}_i^{Scene}$. We investigated the precision of the generated scene text in Section 4.4.2. To further confirm the quality of the obtained scene semantic features (whether different scenes are distinguishable in the feature space), we conducted a visualization of the scene semantic features.

The scene-dependent dataset NWPU provides scene labels for each video, so we visualized the scene semantic features separately for the NWPU training and test sets. As shown in Figure 6, it can be observed that most points corresponding to the same scene cluster together and are separated from points corresponding to different scenes. This indicates that the generated scene semantic features can effectively distinguish between scenes, thus addressing the scene-dependent video anomaly detection task.

## 4.5 Comparison to State-of-the-Art

Finally, we compared our proposed method with the state-of-the-art (SOTA) weakly supervised VAD methods. Since existing weakly

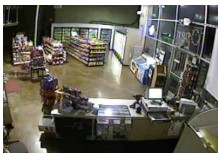 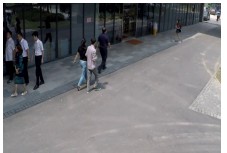 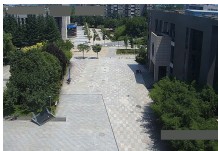 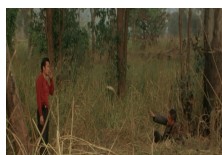 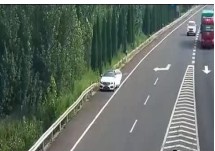

**Generated Scene Text**: The video footage could include the following scenes: *general_store*, *supermarket*, *candy_store*, *department_store*, *gas_station*.

**Generated Scene Text**: The video footage could include the following scenes: *office_building*, *corridor*, *campus*, *food_court*, *cafeteria*.

**Generated Scene Text**: The video footage could include the following scenes: *campus*, *office_building*, *art_school*, *conference_center*, *schoolhouse*.

**Generated Scene Text**: The video footage could include the following scenes: *rice_paddy*, *bamboo_forest*, *badlands*, *rainforest*, *broadleaf*.

**Generated Scene Text**: The video footage could include the following scenes: *highway*, *runway*, *field_road*, *driveway*, *raceway*.

**Figure 7: Visualization results of textual scene descriptions in the CSI module, in which the images are from the UCF_SHT, NWPU, XD-Violence, and TAD datasets. We use the top-5 scene categories as a general description of the scene.**

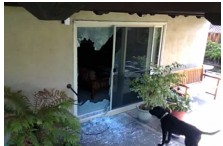 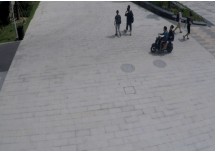 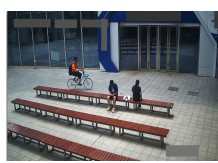 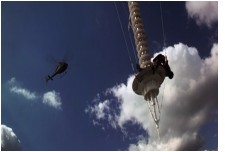 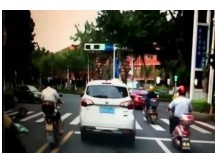

**Generated Object Text**: The video footage could include the following objects: *sliding_door*, *Rottweiler*, *window_screen*, *Labrador_retriever*, *Doberman*.

**Generated Object Text**: The video footage could include the following objects: *motor_scooter*, *moped*, *bicycle-built-for-two*, *tricycle*, *unicycle*.

**Generated Object Text**: The video footage could include the following objects: *park_bench*, *bicycle-built-for-two*, *folding_chair*, *manhole_cover*, *plastic_bag*.

**Generated Object Text**: The video footage could include the following objects: *crane*, *fly*, *airliner*, *coil*, *beacon*.

**Generated Object Text**: The video footage could include the following objects: *traffic_light*, *motor_scooter*, *street_sign*, *passenger_car*, *car_wheel*.

**Figure 8: Visualization results of textual object descriptions in the OSI module, which the images are from the UCF_SHT, NWPU, XD-Violence, and TAD datasets. We use the top-5 scene categories as object descriptions of the scene.**

**Table 5: The comparison of AUC(%) performance on UCF_SHT, NWPU, and TAD, and AP (%) performance on XD-Violence (XD) with SOTA methods. ∗ indicates that the result is reported from [25, 42], and † indicates that the results are obtained by re-training the official codes with different input modalities. The best result under the same settings is bolded and the second best is underlined.**

| Method | UCF_SHT (%) | NWPU (%) | XD (%) | TAD (%) |
|---|---|---|---|---|
| Sultani *et al.* [35] | 77.85† | 65.47† | 75.68* | 81.42* |
| RTFM [42] | 81.32† | 70.57† | 77.81 | 88.15† |
| S3R [46] | - | - | 80.26 | - |
| MGFN [5] | 81.45† | 70.39† | 79.19 | 88.30† |
| UR-DMU [56] | 80.53† | 70.46† | 81.66 | 89.16† |
| Cho *et al.* [6] | - | - | 81.30 | - |
| UMIL [25] | 81.63† | 72.18† | 81.80† | 92.93 |
| Zhang *et al.* [52] | - | - | 81.43 | 91.66 |
| TEVAD [4] | - | - | 79.80 | - |
| CLIP-TSA [14] | - | - | 82.19 | - |
| VadCLIP [48] | 82.15† | 72.64† | 84.51 | 92.70† |
| Ours | **85.94** | **80.22** | **84.69** | **93.90** |

supervised approaches have not been tested on scene-dependent datasets, we retrained and tested these methods on two scene-dependent datasets using publicly available code from these works, attempting to keep the original settings as consistent as possible during training. Given that our model was trained for 50 epochs, to ensure a fair comparison, these retrained models were trained for at least 50 epochs. The performance results are shown in Table 5.

From the comparison results on scene-dependent datasets, our method achieved the best performance on both datasets, indicating that our model can effectively detect scene-dependent anomalies.

Furthermore, compared to other SOTA methods, the performance improvement of our model on both scene-dependent datasets is significant. For instance, compared to the UR-DMU [56], our model improved performance from 80.53% to 85.94% on the UCF_SHT dataset and from 70.46% to 80.22% on the NWPU dataset. It is worth noting that compared to VadCLIP [48], which also utilizes CLIP, our model shows a significant improvement. For example, on the NWPU dataset, our model achieved a performance improvement of 7.58% compared to VadCLIP. For the scene-agnostic datasets XD-Violence and TAD, our proposed method also achieved the best performance. On the TAD dataset, our method surpasses by ∼1% AUC compared to the second-best performing method, UMIL [25], which also utilizes CLIP. Taking into account the results above, our method is effective not only on scene-dependent datasets but also achieves good performance on scene-agnostic datasets.

## 5 CONCLUSION

In this work, we propose a text-driven scene-decouple (TDSD) framework to address weakly supervised video anomaly detection (VAD). By utilizing CLIP to decouple scenes and inject semantic features of scenes and objects separately into the model, we endow the model with the capability to handle scene-dependent anomalies. This paper is also the first work to address scene-dependent weakly supervised VAD. To better evaluate our proposed framework, we reorganized the scene-dependent dataset NWPU to suit a weakly supervised setting. Additionally, we merged the UCF-Crime and ShanghaiTech datasets into the scene-dependent UCF_SHT dataset to facilitate a more comprehensive evaluation. Experimental results demonstrate that our approach achieves significant improvements compared to other methods on two scene-dependent datasets, UCF_SHT and NWPU. Furthermore, it achieves the best performance on two scene-agnostic datasets, XD-Violence and TAD.

# ACKNOWLEDGMENTS

This research was funded by Zhejiang Province Pioneer Research and Development Project "Research on Multi-modal Traffic Accident Holographic Restoration and Scene Database Construction Based on Vehicle-cloud Intersection" (Grant No. 2024C01017).

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
