# OpenReview forum: "TDSD: Text-Driven Scene-Decoupled Weakly Supervised Video Anomaly Detection"
_acmmm.org/ACMMM/2024/Conference — MM2024 Poster_

### Official Review · Reviewer_dNoS · 2024-05-16

**Rating:** 4
**Confidence:** 4

**Summary:**

The manuscript mentioned that many existing methods currently ignore the impact of scenes on anomaly detection. The author believes that scene context plays a crucial role in judging anomalies. The manuscript proposes a novel text-driven scene decoupling (TDSD) framework to solve the challenge of scene-related weakly supervised video anomaly detection by decoupling scenes. The TDSD method consists of TDSD and fine-grained visual enhancement (FVA) modules. The scene decoupling module extracts semantic information from the scene, while the FVA module facilitates fine-grained visual enhancement.

**Strengths:**

The manuscript focuses on the impact of scene information on anomaly detection, which is meaningful for anomaly detection research.
The motivations proposed in the manuscript for using scene and action information are appropriate.
The experiments in the manuscript are relatively sufficient and can also illustrate the effectiveness of the proposed method.

**Limitations:**

The name of the method framework proposed in the manuscript is the same as the module name, which can easily confuse readers.
The content in Figure 2 is very small and not easy to read. It would be better if the content could be made larger. This is just a suggestion. 3.The visualization part shows the effect of effective anomaly detection using scene information and target information. Should some failure examples be shown? 4.The manuscript also lacks discussion of the following latest related work. It is recommended to enrich the writing of related work:

[1] Cloze test helps: Effective video anomaly detection via learning to complete video events[C]//Proceedings of the 28th ACM international conference on multimedia. 2020: 583-591.

[2] Multi-grained Gradual Inference Model for Multimedia Event Extraction[J]. IEEE Transactions on Circuits and Systems for Video Technology, 2024.

[3] Divide-and-Conquer Predictor for Unbiased Scene Graph Generation. IEEE Trans. Circuits Syst. Video Technol. 32(12): 8611-8622 (2022)

[4] A knowledge-based hierarchical causal inference network for video action recognition[J]. IEEE Transactions on Multimedia, 2024.

**Suitability:**

3

---

### Official Review · Reviewer_U4W8 · 2024-05-25

**Rating:** 4
**Confidence:** 3

**Summary:**

The work focuses on scene-dependent anomalies under a weakly supervised setting. The authors construct scene-dependent benchmarks by reorganizing the NWPU dataset and merging UCF-Crime and ShanghaiTech datasets. They propose the Text-Driven Scene-Decoupled Module (TDSD) to inject scene information by leveraging descriptions of both the scene and object, processed with CLIP. Visual features are then enhanced with the proposed Fine-grained Visual Augmentation (FVA). Main and ablated experiments on popular benchmarks demonstrate the effectiveness of the proposed modules, particularly on the curated scene-dependent ones.

**Strengths:**

1. The paper investigates an interesting yet challenging problem of video anomaly detection, where anomalies are scene-dependent.
2. The paper is well-written and easy to follow.
3. The experiments are comprehensive, justifying the module design.
4. Significant gains are obtained in curated scene-dependent datasets including UCF-SHT and NWPU.

**Limitations:**

1. The order of injecting scene/object information is doubtful. Scene information helps object detection and vice versa. Can we swap the order of CSI and OSI.
2. The method is limited by the accuracies of scene/object classification. It would be interesting to see how the performance changes when using a fewer number of scene/object categories in zero-shot classification.

**Suitability:**

3

---

### Official Review · Reviewer_DSqK · 2024-05-28

**Rating:** 2
**Confidence:** 4

**Summary:**

This paper introduces a text-driven scene-decouple framework to address weakly supervised video anomaly detection. The model is enhanced by utilizing CLIP to decouple scenes and  semantic features of scenes and objects to handle scene-dependent anomalies.

**Strengths:**

The paper is well organized and eassy to follow.

**Limitations:**

1. One key of this paper is 'the influence of scenes on anomaly detection', however, the study lacks practical contribution.
For those artificially defined abnormal situations, i. g. bicycles are regarded as anomalies in the ShanghaiTech dataset, while they are regarded as normalities in the Ubnormal dataset, the pre-trained model is lack of generality, because it can’t predict the normal and abnormal events defined by others.  For those common abnormalities, i.g. speeding and fighting that are considered as abnormal events in most scenes, there is essentially no need to study the impact of the scene on such types of anomalies.

2. The original text of Chapter 4.1.1 is ‘ Additionally, we ensure that the anomalies present in the test set are also included in the training set, which is a fundamental assumption in scene-dependent VAD ’. It seems to contradict the original purpose of anomaly detection tasks. Due to the low occurrence probability and diversity of anomaly events, most anomaly detection algorithms have only normal samples for training. Furthermore, test set cannot cover all possible anomalous situations. Therefore, the applicability of a model trained in this way to real-world scenarios is a problem.

3. I understand the decrease in computation speed that arises from employing the CLIP model to augment the precision of anomaly detection. Nonetheless, I think it remains essential to explicate the algorithm's execution speed.

4. The ablation study in this paper does not mention the role of ‘the second CLIP text encoder’ module in CSI and OSI. Can the second CLIP text encoder be replaced with a simpler NLP model to reduce the computational complexity instead of inputting the entire sentence into the entire CLIP text encoder.

**Suitability:**

3

---

### Meta-Review · Area_Chair_1FHr · 2024-07-03

**Recommendation:** Accept (Poster)
**Confidence:** 4

**Metareview:**

This paper introduces a text-driven scene-decouple framework for weakly supervised video anomaly detection. There were mixed ratings initially (two positive and one negative). After the rebuttal, two reviewers were satisfied with the rebuttal, with one improving the rating. The major concern raised by DSqK was not well explained/justified, which can be easily addressed based on existing weakly supervised video anomaly detection frameworks. Hence, the AC agrees with the major opinions on accepting the paper.

---

### Meta-Review · Senior_Area_Chairs · 2024-07-10

**Recommendation:** Accept (Poster)
**Confidence:** 4

**Metareview:**

This paper received mixed ratings initially. After rebuttal, two reviewers tend to accept the paper and one who gave WR did not submit the final rating. SAC and AC carefuuly checked the reviews and rebuttal and recommend acceptance of the paper.